# An IoT-Based GeoData Production System Deployed in a Hospital

**DOI:** 10.3390/s23042086

**Published:** 2023-02-13

**Authors:** Nel Samama, Alexandre Patarot

**Affiliations:** 1SAMOVAR, Télécom SudParis, Institut Polytechnique de Paris, 91120 Palaiseau, France; 2SNPA SAS, Cartobat®, 87200 Saint-Junien, France

**Keywords:** geo-data, indoor positioning, symbolic approach, BLE positioning, hospital

## Abstract

Navigation in large hospitals remains a challenge, especially for patients, visitors and, in some cases, for staff, but in particular it is notable in the case of tracking ambulatory equipment. Current techniques generally seek to reproduce what outdoor navigation systems provide, i.e., “good” accuracy. In many cases, especially in hospitals, reliability is much more important than accuracy. We show that it is possible to realize a simple, reliable system with a low accuracy, but which perfectly fulfills the task assigned in the particular case of tracking stretchers. Optimizing the use of hospital equipment requires the knowledge of its movement. The possibility to access equipment location in real time as well as on the knowledge of the time necessary to move it between two locations allows to predict or to estimate the load and possibly to scale the necessary number of stretchers, and thus the availability of the stretcher bearers. In this paper, an approach of the real-time location of these devices is proposed, and it is called “symbolic”. The principle is described, as well as the practical implementation and the data that can be retrieved. In the second part, an analysis of the results obtained is provided in two directions: the location of stretchers and the determination of travel times. The methodology followed is described, and it is shown that a correct positioning rate of 90% is reached, which is slightly lower than expected, explained by the chosen practical implementation. Moreover, the average error on the determination of travel times is approximately ten seconds on 2 to 7 min trips. The “reliability” (the terminology of which is discussed at the end of the paper) of the results is related to the simplicity of the approach.

## 1. Introduction and Literature Review

In the world of IoT (Internet of Things), which takes place mainly indoors, positioning is, in many cases, particularly important, and it is, in any case, a source of significant performance improvement. GNSS (Global Navigation Satellite System) systems are generally not operational in these environments; therefore, other technical solutions must be proposed. As described by Farahsari [1], many approaches have been explored and some are available to enable location-based services. Liu [2] presents a recent review of the main radio technologies used for indoor positioning, namely WiFi, Bluetooth Low Energy (BLE), Zigbee, RFID (Radio Frequency Identification) or UWB (Ultra Wide Band). The main advantages and disadvantages are described, the important impact of the environment on the performance of all these methods in particular. The review focuses on the objective of obtaining an “accuracy” in “meters” or “centimeters”, i.e., a positioning in a Cartesian form in (x,y,z).

The availability of the position of the numerous pieces of equipment must allow optimizing both their use and their maintenance. In a large structure such as a hospital, some equipment is considered to be “ambulatory”, meaning it is shared by several entities and does not have a specific assigned location. Staff often spend time tracking them down. When this equipment is cheap, the solution is to increase the number of units, but this is not always feasible. The case of the stretchers is investigated in a so-called “departure department” of a hospital. This department manages approximately a hundred stretchers which are intended to supply various entities of the hospital, in particular for examinations such as imaging analyses. The questions in need of an answer concern the possibility to achieve a simple to implement stretcher tracking system that provides reliable geo-location data. In addition, we need to evaluate the real complexity of the infrastructure which is to be deployed. Lastly, there is the need to define the various data which will be available, as well as the usage of these data.

Indeed, our approach resembles geo-fencing approaches which have been proposed by Parise, for instance, in [3] and [4]; however, it is a slightly different in terms of positioning. Indeed, although based on zones (or surfaces), our approach uses all the available measurements of the modules and elaborates a surface with the highest probability of presence (a geometry algorithm is then applied). This resulting surface is then a polynomial that can have various shapes. Moreover, the fact that the mapping of the location is taken into account is again an important difference.

A Data Science approach using recurrent neural networks (RNN) presented by Lukito [5] allows to improve positioning results compared to other data processing methods such as multi-layer perceptron or support vector machine, but not in comparison to more classical methods such as k nearest neighbors. Hoang [6] also discussed the topic of RNNs in the specific case of WiFi not by dealing with positioning, but by applying neural techniques to the trajectory. The results show a significant improvement in the accuracy, even compared to nearest neighbor approaches. Accuracies of less than one meter are reported in many cases, without necessarily using data science techniques; instead, filtering-based approaches are used. Indeed, the measurement noise of radio signals used indoors leads to the need to determine the right approach because the raw processing of measurements leads to results with a high margin of error, from one measurement to the next. Our vision is to consider this variability in the so-called “symbolic” approach described in the following sections, where Alsmadi implements Kalman filters [7] or Shen uses particle filtering [8]. Significant improvements in positioning accuracy are then obtained, ranging from a few centimeters [7] to approximately 1 m [8] after the application of a second filter by assigning weights to the various transmitting beacons, in addition to a Kalman on the measurements.

Garcete addressed the case of BLE [9] by implementing a particle filter in order to obtain an accuracy of 3 m in 74% of the tested situations. The low cost of the solution is highlighted. The latest technological developments easily allow for angle of arrival (AOA) or angle of departure (AOD) measurements of radio signals. These capabilities are also used for positioning, by Abkari, for example [10], in addition to the more traditional time of flight (TOA). When very precise measurements are required, one of the best current techniques is based on TOA measurements in UWB, as proposed by Ho [11], which achieves an average accuracy of approximately 15 cm, paving the way to precise tracking of people, for example. A recent review on UWB has also been made available by Elsanhoury [12].

Radio technologies are not the only ones in the field, as shown by Maheepala [13], where Light Based Indoor Positioning (LIP) systems are exposed in detail, including the main limitations. Li [14] improves the detection of angles in an indoor environment image allowing a significant improvement in the positioning performances. However, the implementation is globally more constrained even if the need for infrastructure is eliminated. A complete review of non-radio-based techniques is proposed by Alam [15], but in the very specific case of “open” spaces, where there are no obstructions to the propagation of signals, which is often an unrealistic hypothesis.

Specifically for hospitals, the same technologies and techniques can be found in various publications. For example, Shipkovenski [16] conducted simulations on a BLE system deployed in a hospital structure with the aim of detecting unauthorized patient discharge as quickly as possible. For Jiangtao [17] using a Zigbee network, the idea is similar in the case of patients suffering from mental pathologies. Some technologies would allow obtaining very good positioning accuracies of a few centimeters at the cost of a large deployment and low distance coverage, as described by Chhetri [18]. This remains reserved for very specific applications. In addition, Anagnostopoulos [19] has conducted a study on the potential uses of a positioning system for the staff of the Geneva hospital.

This global approach of developing a geo-data production tool is more widely inserted in the concept of “smart hospital”. It is, in particular, the framework claimed by Chen [20], who proposes to introduce fuzzy logic to overcome the variability of the measurements. An accuracy of 2 m is offered, taking into account the presence of obstacles and the potential movement of the portable device. The system is based on the use of WiFi signals. Chen [21] also conducts an interesting analysis on wearable devices using WiFi signals from buildings, as well as an implementation on an Arduino platform and indoor positioning in the same context of smart hospitals. In general, the Internet of Things, IoT, is likely capable to provide not only the information on the minute positions of patients, but also a set of connected sensors to monitor them continuously, as reported by Ravali [21]. Let us also mention Lang [22] in order to broaden our perspective; in this particular paper, the modulation of light emitted by LEDs (Light Emitting Diode) is used for both communication and positioning.

Finally, the horizon of a virtual hospital remains the current objective. As discussed by Karakra [23] and Ho [24], this would require a digital twin capable of analyzing and then predicting the real functioning of the hospital and thus optimizing its operation. The approach of locating the various “components” (people and equipment) fits perfectly into this framework.

What we aim to demonstrate is the possibility to implement a simple geolocation system whose performance in terms of accuracy is reduced, but the expectations for the expected use are fully satisfied. The current problem of indoor positioning development is the fact that researchers are focusing on accuracy while the important thing, in many cases, is the reliability of the information provided. Thus, the work is always pushing further the algorithms for processing data that remain, indeed, very noisy. The sources of noise being multiple and very difficult to separate, in particular for systems based on the use of radio signals, the task is still not complete at present. Big data or artificial intelligence techniques, as well as signal processing techniques used a few years ago, still do not provide sufficiently satisfactory results.

Our approach differs from the norm by refocusing on the simplicity of deployment of the system as well as on its reliability. This is true for the so-called symbolic technique that we describe and whose performance we estimate in the context of tracking stretchers in a hospital.

### 1.1. Our Contributions

In this article, we will

✓describe our global so-called “symbolic positioning” approach (the principles), and✓provide further details about the practical deployment that was performed.

The major contribution of this paper is the comparison between the reality on the ground and the data obtained from the system.

The main objective is thus to evaluate the performance of the proposed system in a given context (the movements of stretchers in the hospital).

It will appear, for example, that deployment is not necessarily continuous in a given space. Of course, where the “network” connectivity is not available, it is not possible to provide an exact position of an object, but the system is still able, always with great reliability, to specify that one is no longer within the covered area. 

It is often difficult to compare reality and system performance especially when moving because it is difficult to replicate test results on the ground. We have developed a protocol that allows us to obtain these results in “position” and in “time” by means of a timestamp synchronized with the time of the server recording the data. It is then possible to make a comparative restitution between the reality and the data measured. 

The ways in which such data can be used are again multiple: in the present case, we will focus on two use cases. The first is simply spatial positioning. The second one concerns the temporal reliability considered on a complete itinerary and the accuracy and reliability which can be obtained compared to the real duration of a trip.

### 1.2. Preliminary Remark

Direct comparison with conventional systems using radio signals, typically Bluetooth or WiFi, is complex because the symbolic approach is quite unique in the basic concept. Thus, such a comparison for usual implementations would rely on an accuracy indicator that is not available here, or on a reliability indicator that is not available in conventional cases. Thus, it was decided to expose the approach and the type of possible results in a spirit of sharing. Moreover, in the case of the symbolic approach, it would be relatively simple to provide an “accuracy”. We could, for example, retain the radius of the obtained area (or the square root of the area for complex areas). Such an approach would also lead to providing real-time information on the estimated accuracy. For the deployments performed for this article in a hospital, the typical value is 5 to 6 m. This result seems poor compared to commonly established values of a few meters, but it is achieved with excellent reliability, no calibration, and a very low-density deployment. Finally, this approach was described for the first time in a European conference in 2009 [25], providing more details on the performance compared with some more conventional methods.

Section 2 details our approach of positioning and Section 3 describes the way the system was deployed. The details of the implementation of the experiments and the data available are provided in Section 4. This section is also dedicated to the analyses of the main results obtained, and conclusion is provided in Section 5, followed by a few perspectives in Section 6.

## 2. Details on the Principle of the Symbolic Positioning

### 2.1. The Positioning Principle

#### 2.1.1. General Presentation of the “Symbolic” Positioning

Our approach to localization is simple in its principle: it consists of a strong coupling between cartography on the one hand and radio measurements on the other. The cartography is captured in the form of objects to which attributes are associated. For example, a room is not a traffic zone but only a destination. This point will be fundamental in the next steps, where we attempt to propose navigation and route calculation.

The Initial idea is to define zones according to BLE reception thresholds. The only basic rule is the following: “if the received power is high, then the tag is nearby”. It is important to take into account an additional rule, namely that “if the received power is low, it does not mean that the tag is necessarily far away”. This second assertion is fundamental and constitutes the basis of the approach: the various zones that will be associated with the various power thresholds must not be exclusive. In its complete version, this so-called “symbolic” approach can be schematically represented in Figure 1. The determination of the size of the zones (here, three zones are associated with two reception level thresholds) also depends on the size of the spaces where the modules are deployed. It is clear from Figure 1 that Zone 1, depicted in deep blue, occupies the entire room (where the module, the red star, is installed) while Zone 2, depicted in orange, has a shape that depends on the layout of the building (hence the importance of the mapping step). The same applies to Zone 3 (light blue).

In a practical case, two reception thresholds were defined: for example, −45 dBm for the first threshold (defining Zone 1, close), then −80 dBm for the second threshold, defining Zone 2. It is important to understand that Zone 2 includes Zone 1. This is because it is possible to receive data at low power while being close (because of an obstacle between the tag and the receiving module called “CartoModule”, for example). Without this overlap, the method does not work. This is illustrated by Figure 2, which defines the shape of Zone 1 and Zone 2, respectively, in a typical case.

To summarize, it can be asserted that the localization engine relies on the following:Zone 1 is limited to the room of the CartoModule.In case the room is large (a corridor, for example), Zone 1 is a circle of limited radius (hence the shape of Zone 1 in Figure 2).Zone 2 penetrates the walls with a radius depending on the number of crossed walls. The resulting Zone 2 can thus have a complex shape.

Zone 3 is the entire reception area of the Bluetooth tag (i.e., here, it is the complete floor level).

Each CartoModule that detects a given tag is divided into zones (1, 2 or 3). Then, the positioning algorithm proceeds to a simple intersection (subsequently called area) of the various zones defined by the various CartoModules. The very principle of including zones for a CartoModule (Zone 3 including Zone 2 which includes Zone 1) means that the intersection is never empty. The resulting area is then a polygon.

At the “system” level, it is possible to schematize the approach as follows (see Figure 3 left): from the RSSI measurements for all the tags detected by all the modules, and after having gathered them on a single server allowing them to be precisely time-stamped, the positioning algorithm is launched. It then allows either to visualize all the tags on a map, or to proceed to specific processing and analysis of behaviors or statistics.

It is also possible to trace the path taken by the data (see Figure 3, right) from the Bluetooth tag to the availability of the data to authorized accesses. The BLE signals are detected by the modules which then transmit all the received power levels (from all the detected tags), via the Internet, to the Cartobat servers which will time-stamp the data, sort them and then make them available to the authorized users via a specific URL. From these data, we obtained the results provided in Section 4 below.

The originality of our approach lies in the fact that the CartoModules are composed of plugs directly connected to the existing electrical installation of the building. Thus, the problem of power supply due to the presence of batteries that must be changed is resolved. In addition, the operation is stable with respect to the power supply of the modules. Moreover, the installation requires no intervention into the construction of the building. A second major contribution is that the proposed positioning is said to be “symbolic”, i.e., it is provided in the form of geographical areas of very high probability of presence, and not in the form of precise coordinates. The result is an area with a very high reliability but whose size will depend on the quality of signals received. This corresponds very well to a complex propagation environment such as that inside buildings: there are a multitude of situations in which the signals will not be those expected, and our symbolic approach allows to easily adapt to this kind of common situations.

The principle of a real deployment is a succession of four main steps, as follows:The realization of the “object-oriented” mapping (each element has its own attributes). This aspect is a fundamental element in the calculation of the object position.The implementation of a set of CartoModules. High density is not necessary because then the building architecture (walls, open spaces, electromagnetic characteristics of the partitions, etc.) would have the most impact on the positioning, which is in opposition to our goal. However, depending on the size of the positioning areas one is trying to obtain, this density remains an important parameter.The recovery of the data from the deployed network, i.e., all the BLE power levels, from the various tags measured by the CartoModules. They generally use the building’s WiFi network to send these data to remote servers.The various possibilities of restitution of the tags’ positions: visualization on a map, availability of raw data, etc.

#### 2.1.2. Radio Modules and Cartography

The locations of the CartoModules are part of the mapping. It is this coupling between knowledge of the location of the radio modules and knowledge of the distribution of the spaces that allows an efficient adaptation of the position estimation. Many algorithmic approaches are then possible: the one used in the hospital is described in the following section. In particular, this coupling between module, cartography and algorithm makes it possible to avoid the need for a system calibration phase.

In this type of radio approach, several architectures are possible: calculation by the tag, the infrastructure, measurements in one direction or another, etc. The classical advantages and disadvantages thus apply. Here, the tags are BLE transmitters (and nothing else), and the CartoModules are BLE receivers and WiFi transmitters. The BLE levels originating from the tags are measured by the CartoModules, aggregated, and then transmitted via WiFi and Internet to servers. The algorithms for calculating the position of the tags are implemented on these servers, allowing to always have the latest version and to test various approaches. The restitution can also take various forms, depending on the needs, and be implemented on the servers and not at the terminal level. Currently, this rendition is generally available on any type of terminal with Internet access.

#### 2.1.3. Details of the Localization Engine

The more classical approaches to BLE/WiFi positioning based on radio calibration, sometimes automatic, provide satisfactory results, but our approach operates on a different basis. It seems to us that another way (than in the form of (x,y,z) coordinates) is possible for a positioning in a disturbed radio environment (walls, multiple reflections, propagation depending on “material” characteristics that are complex to integrate, etc.), which is, above all, very dependent on the real environment (presence of people or equipment, orientation of the tags, physical location of the tags, number of connections, etc.). In particular, this applies to the measures that are currently the simplest to implement, namely RSS (Received Signal Strength). The problem is quite different in the case of good quality time-of-flight measurements, such as those obtained with UWB (Ultra Wide Band) modules [12].

The radio part relies on Bluetooth Low Energy (BLE) transmitters in the form of IoT (Internet of Things) tags or a smartphone. These are the elements that are will be followed and whose positions will be calculated. They can be installed on a stretcher, on a medical instrument or even be worn by a patient or staff.

The components of the tracking system are the following:Mapping;BLE tags (see Section 3.1.1);A network of Bluetooth/WiFi modules called CartoModules (see Section 3.1.2), which is the basis of the deployed local positioning network;Secure remote servers;A set of algorithms to estimate the position of the tags;Software components that allow the restitution of raw data or positions on any kind of terminal.

In this paper, components 2 to 5 are addressed, details are proposed on the practical implementation of the system in a hospital, and the results obtained are discussed.

In a typical case, the calculation of the tag’s location is performed by intersections of the symbolic zones obtained for each CartoModule receiving a signal from the considered tag. Considering the zones are currently characterized by polygons, the resulting intersection can take various shapes. In general, the intersection is limited to the boundaries of the mapping, as shown in Figure 4.

Each zone, such as a room, an office, or a corridor, is described by a polygon, which is a set of points in “geojson” format. When the signal of the tag is received by one or several Plug In modules (acting as a gateway between the BLE scan and the WIFI access points), the scan timestamp is sent to the Cartobat cloud (centralized datacenter that both store the data and process the location algorithm) over the Internet. This timestamp is gathered with three pieces of information: the RSS (radio signal strength), the serial number of the module, and the serial number of the tag.

These raw data are stored in a SQL database through an endpoint available over a REST API written in NodeJS to allow simultaneous access. Independently, an automated script is executed, which can be set up to a 2, 5, or 10 s interval depending on the use case. The trade-off is between the cost of the virtual machine CPU and RAM required for a short interval computation and the necessary update frequency. Typically, following the stretchers for process management does not require a short interval, on the contrary to following people for their safety.

This script computes the following algorithm over the last range of raw data, later called list L, available in the SQL database:

1/Sort and group L by tag;

2/Repeat this algorithm for each tag. 

Now, let us focus on a single tag.

3/For each serial number of modules in L, later indexed with i, obtain its horizontal position, later called P, and floor, later called F, on the map.

4/The maximum RSS is used to identify the floor F.

5/Keep only the modules on this selected floor in a list that is later called «Lfloor».

6/For each module in «Lfloor», create a polygon (parameter N is introduced, indicating the number of sides of this polygon).

If N increases, the computation time increases (but it can be decreased at the expense of the geometrical precision in the following part of the algorithm). This polygon represents a circle set by a center that is the position of the «ith» module, and the radius is determined by the RSS. As proximity is the keystone of our concept, three radii (to represent close, medium and far) are introduced. For example, a value of less than −80 dBm indicates that the object is not considered, a value of −80 dBm < RSS < −60 dBm indicates that the distance is too far, a value of −60 dBm < RSS < −40 dBm is medium and that of more than −40 dBm is close. However, these values depend on the electronics considered.

Then, considering the datasheet of the BLE transceiver, the value is set to 10 m for a distance that is far, 5 m for that which is medium and 1 m for that which is close.

7/For each module in «Lfloor», obtain the zone in the map as a polygon. The list of such polygons is later called «Lzones».

8/Intersect all these polygons together to obtain the «most probable location area» called «MPLA».

9/Compute the surface of intersection of MPLA compared to each zone in «Lzones».

10/The zone with maximum intersection with MPLA is the result. If no intersection is ever obtained, which can happen if the emitter is outside the mapped area, then an «out of zone» result is saved.

By construction, this symbolic algorithm cannot compute a position outside the mapped area. The only resulting error can be between floors, between neighbor rooms or due to a slow computation compared with the real-time position.

Thus, the algorithm implemented for the departure department relies on the usual coupling between mapping and BLE measurements, but only the received powers higher than −80 dBm are retained, and the zone intersection is only applied with the two CartoModules that receive the strongest signals. In practice, without having conducted a specific analysis on this point, this very often reduces the algorithm, causing it to consider only one CartoModule, and thus leads to a single zone centered on it.

## 3. Materials and Implementation

### 3.1. Materials

One of the analyses to be carried out was the determination of the average travel time from the departure room to the different imaging departments. It was necessary to determine the routes followed by the stretcher bearers. After some exchanges and occasional follow-ups, it was possible to define the locations for the modules.

#### 3.1.1. The Tags

The Bluetooth Low Energy transmitters used are quite standard. They are fixed on the structure of the stretcher with simple plastic collars that cannot be dismantled, with the antenna pointing towards the ground. Figure 5 shows such a device and its mounting.

In the case of tracking individuals, visitors, patients or professionals, it is, of course, possible to consider the implementation of various types of objects such as a bracelet, a watch, a pendant or even a smartphone.

The technical specifications of the tags currently in use are as follows: 5 × 5 cm^2^, 1 cm thick, BLE, emission power between −20 and +4 dBm, transmission every 100 ms (10 Hz), 2-year battery life (tested in real conditions).

#### 3.1.2. The “CartoModules”

As explained above, once the routes were determined with the stretcher bearers, the modules were deployed along the typical routes. A photo of such a CartoModule is provided in Figure 6 below.

The module includes an AC/DC converter and an ESPRESSIF ESP32 radio module. The protocol stack is written in C with the ESP open-source framework. The BLE frequency is channel 39 (2480 MHz) and the WiFi frequency is 2412, 2437 or 2462 MHz. The typical WiFi received power is between −80 dBm and −50 dBm, and the BLE received power is typically between −80 dBm and −30 dBm.

#### 3.1.3. The Server Connection

The modules have a double task: to measure the power levels received from the tags, and to transmit these values to an access point for transmission to the servers. This second communication is conducted through WiFi. In this case, the CartoModules are directly connected to a specific channel of the hospital’s wireless LAN (in another implementation, gateways or WiFi/4G routers were used). The network architecture for the present case is the one visible in Figure 7.

The electronics of the CartoModules have been designed to allow various implementations. In many network configurations, it is not possible to obtain fixed connection addresses and it is then necessary to adapt the protocol to the local implementation. In this case, the situation was simplified because the hospital’s IT department was able to allocate a static Internet Protocol (IP) address to the CartoModules; this is a simple configuration.

### 3.2. Use Case Experiments in The Hospital

The experiments conducted are based on a simple methodology:Determination of routes to follow;Recording the positions and the corresponding times of a few tags.

The recording of the position is based on a pre-determination of the names of the various areas of the hospital; such an example is provided in Figure 8 (note that in order not to confuse the terms used, we will now refer to “areas” instead of “zones” to characterize the hospital’s surfaces of interest). This “mapping” has been performed on all the covered surfaces of the hospital, in the “departure” department, in particular, as well as in the various imaging departments. As the latter are not necessarily close to each other, it should be noted that the set of areas selected is a small sub-set of the hospital. Note, in addition, that for confidentiality reasons, the following figures are not using the real floor plans.

For example, an area will be labeled as “Area 1” when the tag enters it. The corresponding time is also noted, accurate to the second.

All this must then allow the link between the “ground truth” recorded and the data automatically recorded in the servers (from the deployed system). The routes have been prepared in advance and are made up of a succession of positions characterizing a route. Figure 9 shows such a path, from A to P. Tables are then filled in; they characterize the various routes carried out (see Figure 10).

The result of the experiments is a set of time-stamped tables of passage areas. In each case, each of the two operators carried two tags (typically one in each pocket, or one in a pocket and one around the neck): this is the equivalent of 16 runs (not completely independent, however). Each route was followed four times at different times of the day. This is important because the activity of the services varies during the day and this can have an impact, in particular, on the propagation of the radio waves used, and thus on the performance of the system. There are now 4 × 16, or 64, recorded and annotated routes.

During the experiments, some paths were constructed again, this time with only two tags: one on each carrier. A total of 9 new trips were then recorded (twice for the two operators who moved together).

A final series of 4 trips was carried out in a random way: the crossed areas did not follow a particular route but mixed all the areas. Here, again, two tags were systematically used.

A total of the equivalent of 90 trips, more or less logical with respect to the reality of the service, is thus available. All the results are recorded in a database. It should be noted that we have chosen to use tags carried by operators rather than mounted on one or more stretchers in order to simplify implementation and reduce the inconvenience caused to the departure department. The diversity of the “carriers” is a useful point for a fine understanding of the system behavior. It should also be noted that no significant difference in operation was detected depending on the position of the tag on the operator (this remains an interesting first conclusion).

Our system automatically records the power levels received by the modules (from the tags) and retransmitted by the latter to the servers (via the Internet). The servers are thus continuously fed with such data. Several algorithms (the principles of which are described in Section 2) are then applied to determine the most probable area. For each tag identifier (BLE MAC address of the tag), a file is then providing the corresponding area for each instant when a measurement is available. An example of this file is deminstrated in Figure 11.

As can be seen, an additional indicator, “fiabilite” (which means “reliability”), is also available, in addition to the GPS (Global Navigation System)-compatible Longitude and Latitude values. This indicator is quite important since it can be used in order to apply weight values to the various CartoModule measurements and hence have an impact on the positioning algorithms.

## 4. Results and Analyses

The needs of the hospital’s departure department are to be able to locate all of its stretchers in real time in order to be able to repatriate some of them efficiently, if necessary, but also to allow statistical analyses of the travel times to and from the department and the imaging departments. This second aspect of research has the potential, in the longer term, of providing data that will allow to estimate the probability of a shortage of stretchers according to the criteria of affluence or of being able to predict the availability of stretchers.

Two types of comparisons are made between real data and data recorded and calculated by the system:In order to determine the detection efficiency of the tag areas;In order to estimate the accuracy of determining the duration of a journey.

### 4.1. Probability of Presence in a Given Area

The principle of the validation is relatively simple: from an entry in the ground truth table for a given tag (identified by its MAC address in the “server” file), the time of reading is used. The algorithm searches for this MAC address in the “server” file, then examines the data provided by the area determination, and compares it with the real area recorded during the experiments. This process is repeated for all 90 surveys, distinguishing between four types of routes (from the departure room to the four most frequent destinations).

As can be seen in Figure 9, some points are not in any predetermined area (as is the case of point K in Figure 9), and the problem of conducting the correspondence arises. The same is true for points that are located at the border between two adjacent areas (as is the case of points C or L in Figure 9). Thus, a color code has been established: green is assigned to the areas that match perfectly, orange to those that are adjacent and red to those that do not match.

Table 1 provides an example of results for a given run. The table shows all the points tested during a round trip (rows) for one tag in this example (columns). The color code used makes it possible to very quickly visually realize the global performance (Figure 12 is such an example for several tags).

Figure 12 shows a set of area determination results for four identical trips (the four groups of points numbered from A to O) conducted at different times and under different conditions (corridor congestion, elevator availability, etc.). The colored columns correspond to four different tags (identifiers in the first row), carried in pockets or around the neck. The simple color code allows to quickly characterize the performance: red for a detected area error, orange for an adjacent area, and green for the right area.

Table 2 provides a summary of all the paths performed. A total of 90% of the areas were correctly estimated (green or orange). This result is not so satisfactory because the reliability of the system is usually much higher than that (rather 98 or 99%).

The reason for this relative underperformance lies in the poor distribution of the modules in the building. In particular, it is disruptive for the symbolic positioning to place modules one above the other on different floors. The issue of optimization of the deployments is thus a topic for work to come. By simply removing these few situations, a reliability of 97% is reached, in line with the real capabilities of this positioning system.

### 4.2. Discussion Concerning the Positioning Accuracy

A recurring question concerns the positioning accuracy that can be obtained with this positioning method. Let us reiterate here that this parameter is clearly not the one that is sought in the symbolic approach and that the deployment carried out presents a low density of modules considering the surface covered (26 modules for nearly 7000 m^2^). However, the question deserves to be addressed. To do so, we propose to compare two methods of calculating this accuracy, as follows:⮚The first one consists in defining as position the center of the determined symbolic area (the red points in Figure 13), then measuring the true distance, in meters, between the true position and this center (column “Symbolic equivalent accuracy” in Table 3);⮚The second one is based on the determination of the module which receives the tag with the highest power (column “Nearest module accuracy” in the table). This approach remains compatible with the “symbolic” principle, but does not realize the zone intersections described in the algorithm. As a result, it turns out that the reliability of this approach is degraded.

The results provided in Table 3 show, on the one hand, that the notion of accuracy is not well adapted to our approach if we consider the resulting areas as references, but also that it is still possible, with a symbolic approach, to provide an accuracy indicator of a few meters (“nearest modulus” approach). However, it is important to note that this last approach does not provide the same reliability as the initial symbolic approach. Moreover, these results were obtained with 26 modules deployed to cover an area of 6850 m^2^. This represents a density of one module per 260 m^2^ (typically a 16 m × 16 m square).

### 4.3. Accuracy of the Travel Time Estimation

As a corollary, but according to an identical principle, it is possible to estimate the accuracy of the travel time from the same data. It is then a question of comparing the time actually taken to move from the departure department to one of the other imaging departments with the system time between the departure from the departure department (this is the practical difficulty) and the arrival at the imaging department. The problem is twofold: we need to automatically determine the departure from the department and define the arrival at the destination department.

The following procedure is applied:-Departure is defined as the moment when the system detects the exit from the initial area (that of point A). In some cases, as this point A is not in fact always in a well-defined area, an error is generated at the beginning of the estimation;-The arrival, which is easier to define, corresponds to the first detection of the destination area.

It is clear that in the future, it will be necessary to be able to define these various areas more precisely.

These analyses were conducted for all 90 routes, and the following figures present the synthesis of the results in the form of a detailed result for one destination and one tag (Table 4). This summary shows accuracy of the order of approximately 15 seconds on runs of a typical duration of 2 to 3 min.

The typical travel times from the departure service to the other services fall in the range between 2 and 7 min. Considering the case depicted in Table 5, the average duration of the return travel is slightly shorter than that of the forward travel. Moreover, one can observe that there is a real difference in the amplitude between the quickest and the slowest forward travel time and the same travels for the return path.

These results show that it is possible both to obtain an average value of the displacements with a precision of about ten seconds, but also to follow, in real time, the disparity of the movements of the stretchers. It is the combination of these two aspects of the problem that leads us to qualify the system as “reliable”. Thus, there is a way, other than the absolute search for positioning accuracy (which remains a relevant solution, of course), to obtain useful geo-localized data.

## 5. Conclusions

After presenting the principles of the proposed symbolic approach, a description of the required components is provided. The deployment in a hospital for an emergency service is then detailed with the two main use cases: tracking stretchers and determining various travel times. The results show a 90% reliability of good area detection, as well as a capacity to estimate travel times of approximately 15 s for trips lasing between 3 and 7 min.

These results show that the proposed symbolic approach allows the extraction of indicators of good quality services. The fine precision of the measurements is not sought, but it is sufficient to meet the objectives of a reliable estimation of trips and travel times. The approach is simple but reliable and efficient, at least for the present purpose.

The future objectives are of a variety of natures. For a hospital, it is a question of developing uses based on these geo-localized data. As shown in the bibliography, there are some thoughts in this field, and in particular in the context of “digital twins”. For the technical system presented, it is a question of continuing the developments in order to further simplify the deployments. Work has been conducted in recent years on a simple and fast mapping tool. It is necessary now to put in place an installation procedure that allows local building management teams to deploy the system. The management of the “network” aspects is part of these thoughts.

## 6. Perspectives

From an academic point of view, we intend to continue to work on calculation algorithms (based on signal measurements), but we are also interested in the “data processing” aspects. The data generated by the current system could be used in order to propose relevant indicators, such as, for example, indicators of a person’s activity or the use of equipment.

## Figures and Tables

**Figure 1 sensors-23-02086-f001:**
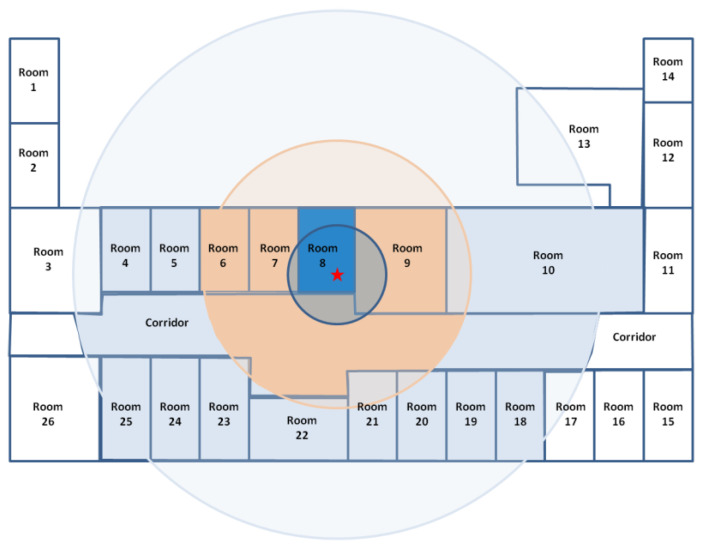
Philosophy of the symbolic positioning.

**Figure 2 sensors-23-02086-f002:**
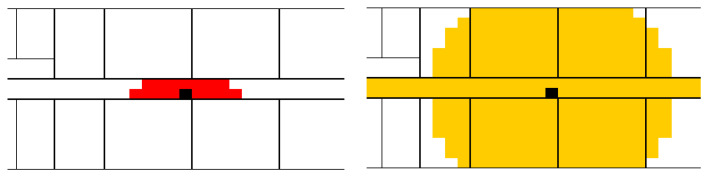
Zone 1 (in red) for the module in black, located in the middle of the corridor, and Zone 2 (in yellow), overlapping.

**Figure 3 sensors-23-02086-f003:**
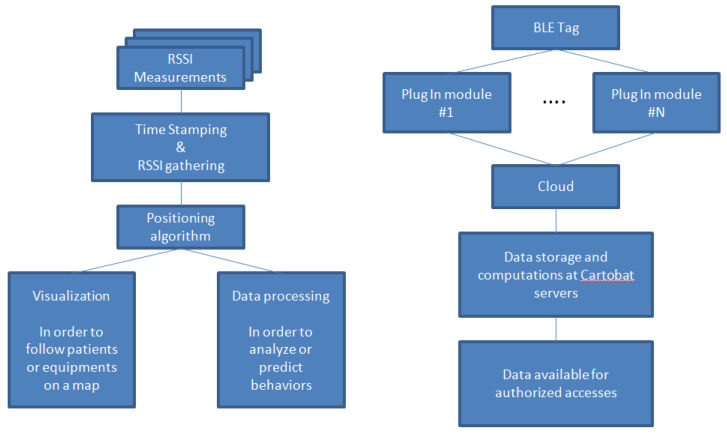
System model diagram (**left**) and System flow chart (**right**).

**Figure 4 sensors-23-02086-f004:**
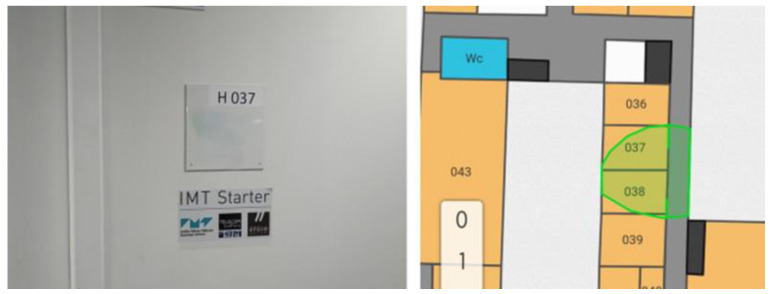
The principle of coupling mapping and location: the green surface is the resulting location which does not require any calibration or model design. On the left is the “visual” location of the tag, and on the right is the area calculated by the system (in green).

**Figure 5 sensors-23-02086-f005:**
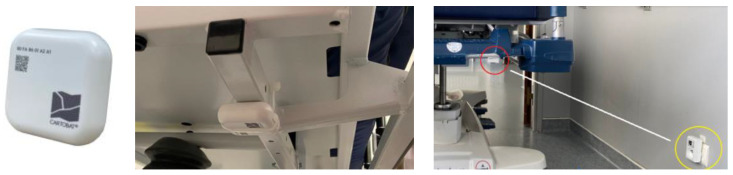
A typical tag mounted on a stretcher. The yellow circle in the right figure shows a plugged CartoModule.

**Figure 6 sensors-23-02086-f006:**
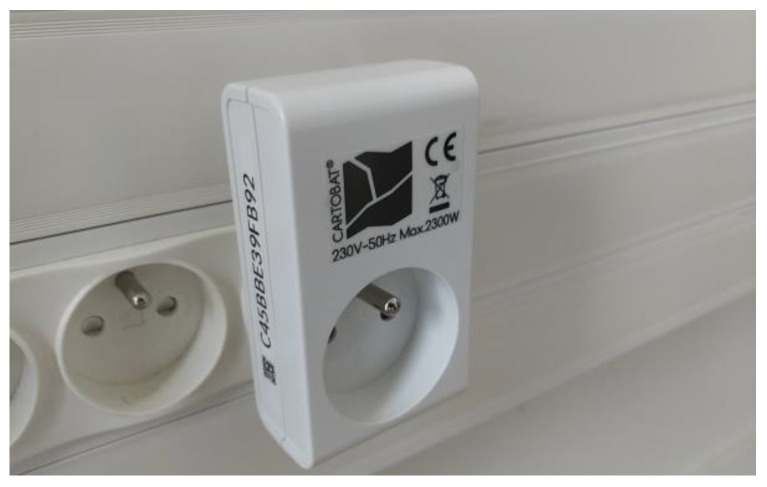
A plugged CartoModule.

**Figure 7 sensors-23-02086-f007:**
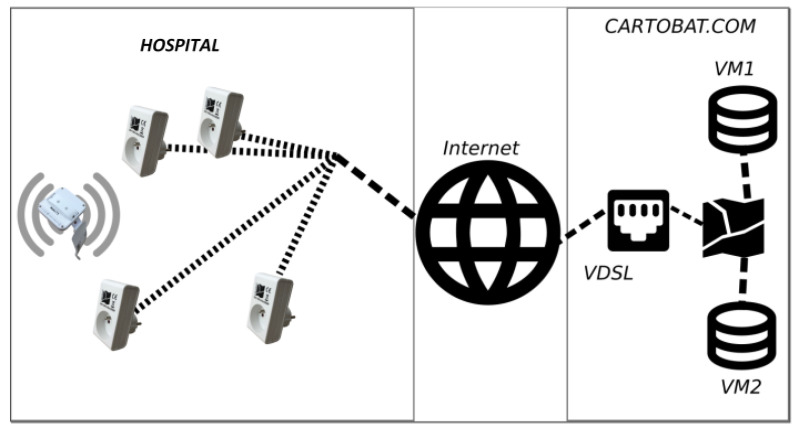
Global network architecture used in the hospital.

**Figure 8 sensors-23-02086-f008:**
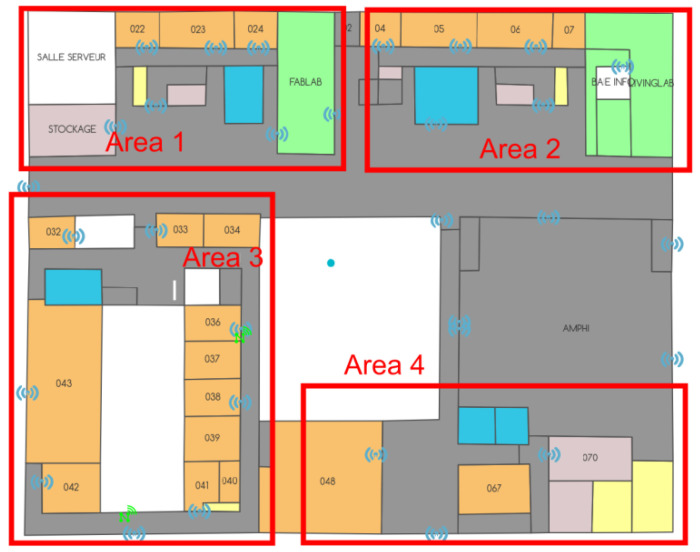
“Area naming”.

**Figure 9 sensors-23-02086-f009:**
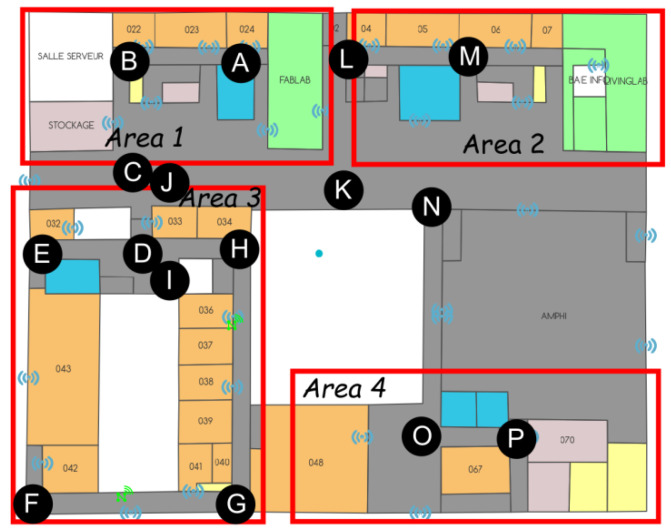
Definition of a “path”, in this case, from “A” to “P”.

**Figure 10 sensors-23-02086-f010:**
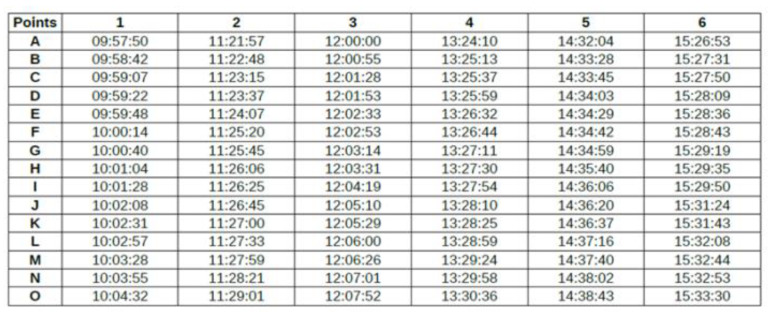
A screen capture of a typical “experiment file” from the departure service to an imaging service (all the data are time in the hh:mm:ss format).

**Figure 11 sensors-23-02086-f011:**
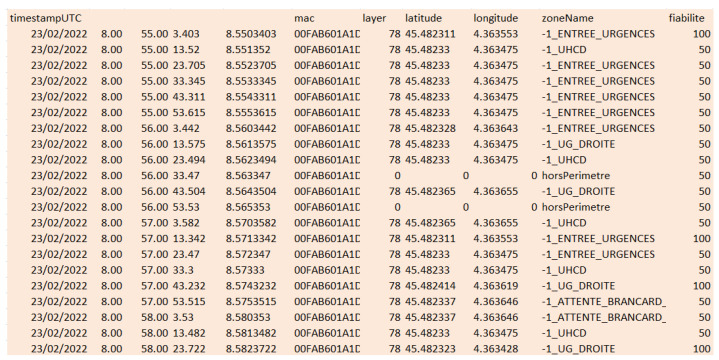
Detail of typical available data.

**Figure 12 sensors-23-02086-f012:**
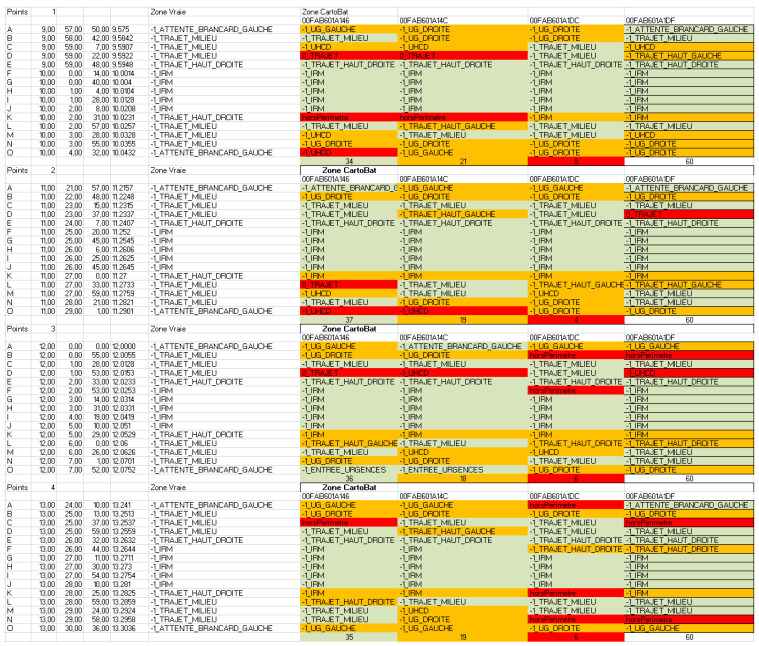
Analyses file for positioning reliability for four tags and four destinations.

**Figure 13 sensors-23-02086-f013:**
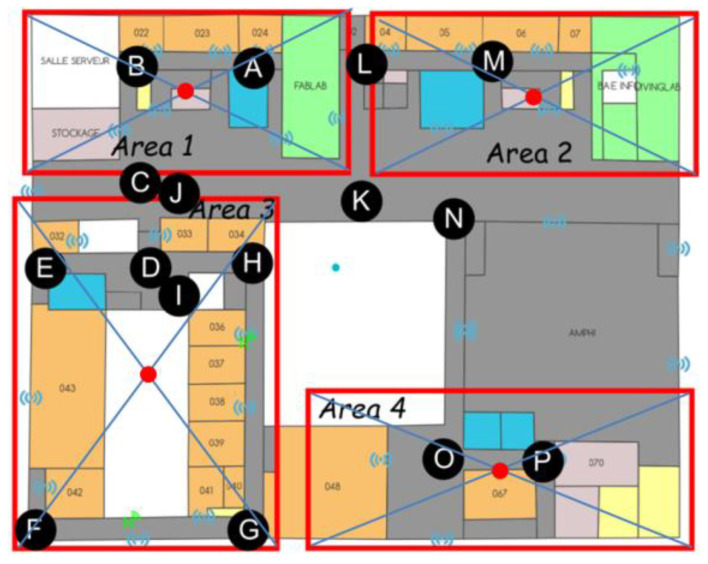
Definition of the central points for any given area (in red).

**Table 1 sensors-23-02086-t001:** Analyses file for positioning reliability for one tag and one destination.

Points	Time	Real Location	Calculated Location
			A1CC
A	10:25:23	-1_ATTENTE_BRANCARD_GAUCHE	-1_UG_GAUCHE
B	10:25:45	-1_UG_GAUCHE	-1_UG_GAUCHE
C	10:26:33	-1_UG_DROITE	-1_UG_DROITE
D	10:27:08	-1_TRAJET_MILIEU	-1_TRAJET_MILIEU
E	10:27:49	-1_TRAJET_HAUT_DROITE	-1_TRAJET_HAUT_DROITE
F	10:28:34	-1_IRM	-1_IRM
G	10:29:28	-1_IRM	-1_IRM
H	10:30:11	-1_TRAJET_HAUT_DROITE	-1_TRAJET_HAUT_DROITE
I	10:30:39	-1_TRAJET_MILIEU	-1_TRAJET_MILIEU
J	10:30:39	-1_TRAJET_MILIEU	-1_TRAJET_MILIEU
K	10:31:19	-1_UG_DROITE	-1_UG_DROITE
L	10:31:54	-1_UG_GAUCHE	-1_UG_GAUCHE
M	10:32:14	-1_ATTENTE_BRANCARD_GAUCHE	-1_ATTENTE_BRANCARD_GAUCHE
N	10:32:56	-1_UG_DROITE	-1_UG_DROITE
O	10:33:25	-1_UG_GAUCHE	-1_UG_GAUCHE
P	10:34:08	-1_ATTENTE_BRANCARD_GAUCHE	-1_ATTENTE_BRANCARD_GAUCHE

**Table 2 sensors-23-02086-t002:** Positioning determination accuracy.

Destination Service		Summary Results			
		Green	Orange	Red	Σ
Imaging department 1	Nb points	190	98	12	300
	%	63	33	4	100
	Cumulative %	→	96	100	100
Imaging department 2	Nb points	176	82	42	300
	%	59	27	14	100
	Cumulative %	→	86	100	100
Imaging department 3	Nb points	273	75	52	400
	%	68	19	13	100
	Cumulative %	→	87	100	100
Imaging department 4	Nb points	350	131	39	520
	%	67	25	8	100
	Cumulative %	→	93	100	100
Σ	Nb points	989	386	145	1520
	%	65	25	10	100
	Cumulative %	→	90	100	100

**Table 3 sensors-23-02086-t003:** Accuracy of the proposed approach with respect to the applied positioning method.

Points	Time	Calculated Location	Symbolic Equivalent Accuracy	Nearest Module Accuracy
		A1CC	(m)	(m)
A	10:25:23	-1_UG_GAUCHE	16.0	3.5
B	10:25:45	-1_UG_GAUCHE	16.0	3.5
C	10:26:33	-1_UG_DROITE	17.0	6.0
D	10:27:08	-1_TRAJET_MILIEU	4.5	3.0
E	10:27:49	-1_TRAJET_HAUT_DROITE	22.0	5.0
F	10:28:34	-1_IRM	1.0	6.0
G	10:29:28	-1_IRM	25.0	6.0
H	10:30:11	-1_TRAJET_HAUT_DROITE	12.5	4.0
I	10:30:39	-1_TRAJET_MILIEU	17.5	2.0
J	10:30:39	-1_TRAJET_MILIEU	12.5	3.0
K	10:31:19	-1_UG_DROITE	25.0	5.0
L	10:31:54	-1_UG_GAUCHE	1.0	4.0
M	10:32:14	-1_ATTENTE_BRANCARD_GAUCHE	22.0	5.0
N	10:32:56	-1_UG_DROITE	4.5	2.5
O	10:33:25	-1_UG_GAUCHE	16.0	6.0
P	10:34:08	-1_ATTENTE_BRANCARD_GAUCHE	7.5	6.0
* **Average resulting accuracy** *			13.75	4.41
* **Standard deviation** *			8.03	1.39

**Table 4 sensors-23-02086-t004:** Travel time comparison for a given imaging service.

Points	Time	Real Location	Travel Time
			00FAB601A146
A	09:57:50	-1_ATTENTE_BRANCARD_GAUCHE	Real forward travel
B	09:58:42	-1_TRAJET_MILIEU	00:02:24
C	09:59:07	-1_TRAJET_MILIEU	Measured forward travel
D	09:59:22	-1_TRAJET_MILIEU	00:02:20
E	09:59:48	-1_TRAJET_HAUT_DROITE	Error
F	10:00:14	-1_IRM	00:00:04
G	10:00:40	-1_IRM	
H	10:01:04	-1_IRM	
I	10:01:28	-1_IRM	
J	10:02:08	-1_IRM	Real return travel
K	10:02:31	-1_TRAJET_HAUT_DROITE	00:02:24
L	10:02:57	-1_TRAJET_MILIEU	Measeured return travel
M	10:03:28	-1_TRAJET_MILIEU	00:02:30
N	10:03:55	-1_TRAJET_MILIEU	Error
O	10:04:32	-1_ATTENTE_BRANCARD_GAUCHE	00:00:06

**Table 5 sensors-23-02086-t005:** Travel time accuracy.

Nb of Paths	20			
	Forward Travel Time (min)		Return Travel Time (min)	
Real travel	Min	00:01:50	Min	00:02:06
	Max	00:03:23	Max	00:02:42
	Average	00:02:42	Average	00:02:26
Measured travel	Min	00:01:40	Min	00:01:50
	Max	00:03:00	Max	00:03:00
	Average	00:02:29	Average	00:02:22
Mean error (s)	15		13	

## Data Availability

Data sharing is not applicable to this article.

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
