# Peer review of "An IoT-Based GeoData Production System Deployed in a Hospital"

_sensors, 2023, doi:10.3390/s23042086_

Round 1

Reviewer 1 Report

The proposed technique is not clear

The positioning methodology is not presented in a scientific manner

the Positioning methodology does not have a mathematical presentation or proof 

No validation of the proposed technique is presented 

If the model is correct it will be beneficial in its topic.

The authors have to illustrate the model in more details I think it will be more accurate if they add the following parts

-System model diagrams

-System flow chart

-Mathematical formulation of the proposed technique

-validation and comparison with other techniques that addressing the same question

Author Response

First of all we would like to thank the reviewers for their suggestions for improvement and their comments. We detail in this document the responses we make and how we modified the original paper. We believe that this new version is indeed much better: more readable and in line with the expectations of potential readers.

REVIEWER 1

The proposed technique is not clear

The positioning methodology is not presented in a scientific manner the Positioning methodology does not have a mathematical presentation or proof

No validation of the proposed technique is presented

If the model is correct it will be beneficial in its topic.

As discussed hereafter, a new architecture of the paper has been defined, including the suggested details, together with some additional discussions along the original text (in red in the new uploaded document).

The authors have to illustrate the model in more details I think it will be more accurate if they add the following parts

-System model diagrams Added in a new chapter 2 “Details on the principle of the symbolic positioning” –  See diagram below

-System flow chart Added in a new chapter 2 “Details on the principle of the symbolic positioning” –  See flow chart below

-Mathematical formulation of the proposed technique Added in a new chapter 2 “Details on the principle of the symbolic positioning” –  See algorithm below

-validation and comparison with other techniques that addressing the same question A new section has been added (“Preliminary remark”) following “our contributions” in order to explain the difficulty to achieve such a goal. It also sent back to a former publication on the symbolic positioning.

System model diagram and System flow chart

Mathematical formulation of the proposed technique

Initialization

  • “i” is associated with the Tags => Tags(i)
  • “j” is associated with the plug-in modules => PlugIn(j)
  • T1 and T2 are the two power level thresholds defined once for all for all the Tags and all the Plug-In modules
  • For each Tag(i)
    • Measurement of the RSSI(i,j) measured by PlugIn(j) from Tag(i)
    • Determination of k(i,j) = 1,2,3 depending of the RSSI value wrt the two thresholds T1 and T2.
      • k(i,j) = 1 if RSSI(i,j)>T1
      • k(i,j) = 2 if T2<RSSI(i,j)<T1
      • k(i,j) = 3 if RSSI(i,j)<T2
    • For each PlugIn(j)
      • Calculation of the three corresponding areas Area(j, k(i,j))
        • Area(j,1)
        • Area(j,2)
        • Area(j,3)

Positioning

  • For a given Tag(i)
    • For all the PlugIn(j)
      • Obtaining Area(j, k(i,j))
    • Calculation of ResultArea(i) =

There are different possible approaches in order to obtain the Area(j, k(i,j)). We present below two algorithms. In the present paper, algorithm 2 has been used. Note that it assumes that the map, including the locations of all the Plug-In modules and all the symbolic areas (rooms, corridors, walls, etc), is available.

Algorithm 1

  • Area(j,1) = area including PlugIn(j)
  • Area(j,2) = Area(j,1) + adjacent areas
  • Area(j,3) = Area(j,1) + adjacent areas + adjacent areas of adjacent areas

    = Area(j,2) + adjacent areas

Please refer to [IAIN2009] for the full definition of “adjacent areas”.

Algorithm 2

  • Area(j,1) = circle of center PlugIn(j) and radius R1
  • Area(j,2) = circle of center PlugIn(j) and radius R2
  • Area(j,3) = circle of center PlugIn(j) and radius R3

R1, R2 and R3 being defined once for all, for all the modules and all the tags. Typical values are 4 meters, 8 meters and 16 meters.

Reviewer 2 Report

In terms of positioning accuracy, the performance of proposed method is poor, but the presented application does not need high positioning performance, and the performance of proposed method is sufficient for the application. On the other hand, it is very meaningful to position and manage stretchers in the whole hospital, especially for hospitals with limited resources. Moreover, it can also be extended to the positioning management of more people and things in the future, so as to realize the optimal allocation of other kinds of resources.

The writing should be improved significantly.

Author Response

First of all we would like to thank the reviewers for their suggestions for improvement and their comments. We detail in this document the responses we make and how we modified the original paper. We believe that this new version is indeed much better: more readable and in line with the expectations of potential readers.

REVIEWER 2

In terms of positioning accuracy, the performance of proposed method is poor, but the presented application does not need high positioning performance, and the performance of proposed method is sufficient for the application. On the other hand, it is very meaningful to position and manage stretchers in the whole hospital, especially for hospitals with limited resources. Moreover, it can also be extended to the positioning management of more people and things in the future, so as to realize the optimal allocation of other kinds of resources.

Thanks for these positive comments. We agree.

The writing should be improved significantly.

A proofreading was carried out and some corrections applied.

Reviewer 3 Report

Comments to Author

The author proposed symbolic positioning, the real-time localization of hospital equipment (e.g., the stretcher bearers). The practical implementation has been explained and the analysis of the results obtained is given in two directions: the localization of stretchers and the determination of travel times. The idea of the paper is good, but author fails to present the contribution and the system design. My comments to enhance the paper as follows:

1) Abstract should clearly mention the challenge of the positioning in general and in the hospital. It also should include brief of how the authors will solve the problem.  

2) The challenges/problem statement should be explained before the contribution.

3) The contribution is not clear. I suggest writing it in specific points. The motivation/ objectives can also be written before contribution.

4) I recommend adding a complete section about the system design which includes flowchart and architecture diagrams, and algorithm steps. It should be added before the implementation and the deployment.

5)  The conclusion is not available. Authors should add it.

6) References are not sufficient. Authors are recommended to add relevant and recent references.

Author Response

First of all we would like to thank the reviewers for their suggestions for improvement and their comments. We detail in this document the responses we make and how we modified the original paper. We believe that this new version is indeed much better: more readable and in line with the expectations of potential readers.

REVIEWER 3

The author proposed symbolic positioning, the real-time localization of hospital equipment (e.g., the stretcher bearers). The practical implementation has been explained and the analysis of the results obtained is given in two directions: the localization of stretchers and the determination of travel times. The idea of the paper is good, but author fails to present the contribution and the system design.

My comments to enhance the paper as follows:

1) Abstract should clearly mention the challenge of the positioning in general and in the hospital. It also should include brief of how the authors will solve the problem. 

A few sentences have been added in the abstract.

2) The challenges/problem statement should be explained before the contribution.

Two paragraphs have been added just before the section “Our contributions” in order to explain the “starting point” of our work.

3) The contribution is not clear. I suggest writing it in specific points. The motivation/ objectives can also be written before contribution.

The part “Our contributions” has been rearranged accordingly.

4) I recommend adding a complete section about the system design which includes flowchart and architecture diagrams, and algorithm steps. It should be added before the implementation and the deployment.

The architecture of the paper has been changed. The new chapter 2 hopefully answers your remark.

5)  The conclusion is not available. Authors should add it.

The former “Discussion” is indeed the conclusion of this work. It has been changed.

6) References are not sufficient. Authors are recommended to add relevant and recent references.

A few additional references have been added, but you comment is a little bit rough since references are rather recent and, from our point of view, quite relevant.

Reviewer 4 Report

The results are mainly based on Figure 8, showing the path, which is not the right one (it is the same as Figure 7). Thus, the evaluation of the results is not possible.

Please resubmit with the right figure.

Author Response

First of all we would like to thank the reviewers for their suggestions for improvement and their comments. We detail in this document the responses we make and how we modified the original paper. We believe that this new version is indeed much better: more readable and in line with the expectations of potential readers.

REVIEWER 4

The results are mainly based on Figure 8, showing the path, which is not the right one (it is the same as Figure 7). Thus, the evaluation of the results is not possible.

You are right: thanks for your careful reading of the paper. The right (former) Figure 8 (now Figure 9 in the revised paper) has been included.

Please resubmit with the right figure.

Results will now be understandable.

Round 2

Reviewer 1 Report

The presented Mathematical technique in pages 8 and 9 needs to be more described.

The RSS measurement technique is not descrived.

The error is not calculated and compared to other indoor positioning systems.

Author Response

Please see attached pdf file.

Reviewer 3 Report

Authors addressed all my comments.

Author Response

Authors addressed all my comments.

Thank you for your valuable suggestions and remarks.

Reviewer 4 Report

The authors took into account all the reviewers' comments. I have no further comments.

Author Response

The authors took into account all the reviewers' comments. I have no further comments.

Thank you for your valuable suggestions and remarks.